# *Agaricus blazei*-Based Mushroom Extract Supplementation to Birch Allergic Blood Donors: A Randomized Clinical Trial

**DOI:** 10.3390/nu11102339

**Published:** 2019-10-02

**Authors:** Faiza Mahmood, Geir Hetland, Ivo Nentwich, Mohammad Reza Mirlashari, Reza Ghiasvand, Lise Sofie Haug Nissen-Meyer

**Affiliations:** 1Department of Immunology and Transfusion Medicine, Oslo University Hospital, 0407 Oslo, Norway; faiza.mahmood@ahus.no (F.M.); ivo.nentwich@ous-hf.no (I.N.); uxmoir@ous-hf.no (M.R.M.); lise.sofie.haug.nissen-meyer@ous-hf.no (L.S.H.N.-M.); 2Department of Immunology and Transfusion Medicine, Akershus University Hospital, 1478 Lørenskog, Norway; 3Department of Immunology, Institute of Clinical Medicine, University of Oslo, 0318 Oslo, Norway; 4Department of Biostatistics, Institute of Basic Medical Sciences, University of Oslo, 0372 Oslo, Norway; reza.ghisvand@medisin.uio.no

**Keywords:** *Agaricus blazei* mushroom, Andosan^TM^, asthma, basophil activation, birch pollen allergy, clinical trial

## Abstract

Since *Agaricus blazei* Murill (AbM) extract reduced specific IgE and ameliorated a skewed Th1/Th2 balance in a mouse allergy model, it was tested in blood donors with self-reported, IgE-positive, birch pollen allergy and/or asthma. Sixty recruited donors were randomized in a placebo-controlled, double-blinded study with pre-seasonal, 7-week, oral supplementation with the AbM-based extract Andosan^TM^. Before and after the pollen season, questionnaires were answered for allergic rhino-conjunctivitis, asthma, and medication; serum IgE was measured, and Bet v 1-induced basophil activation was determined by CD63 expression. The reported general allergy and asthma symptoms and medication were significantly reduced in the AbM compared to the placebo group during pollen season. During the season, there was significant reduction in specific IgE anti-Bet v 1 and anti-t3 (birch pollen extract) levels in the AbM compared with the placebo group. While the maximal allergen concentrations needed for eliciting basophil activation before the season, changed significantly in the placebo group to lower concentrations (i.e., enhanced sensitization) after the season, these concentrations remained similar in the Andosan^TM^ AbM extract group. Hence, the prophylactic effect of oral supplementation before the season with the AbM-based Andosan^TM^ extract on aeroallergen-induced allergy was associated with reduced specific IgE levels during the season and basophils becoming less sensitive to allergen activation.

## 1. Introduction

*Agaricus blazei* Murill (AbM) is a Brazilian relative of champignon, and *Hericium erinaceus* and *Grifola frondosa* from Asia are all medicinal *Basidiomycetes* mushrooms used as health food and against cancer [1,2,3]. Their antitumor properties are partly immunomodulatory via stimulation by, e.g., β-glucans of monocytic, natural killer (NK), and T helper 1 cells [4,5,6,7]. The Th1 immune response is reciprocal to the Th2 response, which promotes allergy and asthma [8]. An AbM extract given orally to asthma-induced and tumor-bearing mice ameliorated their skewed Th1/Th2 balance and reduced specific IgE and tumor load [9]. Andosan^TM^ is a Japanese extract of 82% AbM, 15% *H. erinaceus*, and 3% *G. frondosa* that gave anti-allergic effects in a mouse allergy model, as demonstrated by reduced serum IgE and IgG1 anti-ovalbumin levels and shift from Th2 to Th1 predominant cytokine profiles in spleen cell cultures [10]. We have recently found that Andosan^TM^ in drinking water has antitumor effects in a mouse model for colorectal cancer [11]. Thus, both anti-allergic/-asthmatic and antitumor properties have been demonstrated in vivo for AbM given enterally.

Atopic diseases like rhino-conjunctivitis and allergic bronchial asthma affect 20%–25% of the population in developed countries [12], but prophylactic measures have limited effect, particularly on aeroallergy [13]. The aims for both atopy prophylaxis and treatment are improvement of clinical symptoms by reversing Th2-dominant atopic phenotype to nonatopic Th1 phenotype. Immunomodulation by food supplements is a simple treatment for atopy with few side effects, as previously documented for β-glucans on symptoms of pollen allergy [14,15].

Allergic and asthmatic blood donors can give blood if they are not strongly affected and/or have not had anaphylactic reactions or taken corticosteroids. Since donors with pollen allergy represent a select group with regard to symptoms and allowed medication, standard recommendations of clinical outcome for allergic rhino-conjunctivitis [16] cannot be readily used in such a clinical study of pollen-induced allergy. Therefore, in a blood bank setting, we chose to use a modified (in-house) questionnaire that also evaluates asthma. In southeastern Norway, the birch pollen season is from mid-April to mid-May.

Basophilic granulocytes (“basophils”) and tissue mast cells [17] are involved in the immediate type-I allergic reaction, where allergen-specific IgE antibodies in plasma are bound to high-affinity Fcε receptor (R) I (FcεRI) on the effector cells. They are then activated by allergen cross-linking of the bound IgE, which results in degranulation of the cells and release of mediators like histamine and arachidonic acid metabolites, e.g., prostaglandin D_2_ after activation of cyclooxygenase-2 (COX-2), and proinflammatory cytokines, e.g., IL-6 [18,19,20].

Basophil activation test (BAT) is a validated functional test for allergy in which basophils isolated from blood samples of people with allergies are stimulated ex vivo with the particular allergen the individual is allergic to, here, birch pollen allergen. When relevant allergen is added, it will cross-link the FcεRI-bound allergen-specific IgE on the cells and thus activate the basophils to degranulation and COX-2 activation that incite an allergic reaction. This activation can be detected in vitro when cells after allergen co-incubation are analyzed by flow cytometry with fluorochrome-labelled IgG antibodies specific to activation markers such as CD63 on basophil surface [21]. In highly allergic individuals, even very low concentrations of the particular allergen will provoke basophil activation in contrast to the situation for less allergic individuals where higher allergen concentrations are needed. If no activation is achieved regardless of allergen concentration, the individual is not allergic to the allergen being tested. However, ≥5% of clinically allergic individuals have a negative BAT [22], which probably is due to aberrant intracellular signaling after FcεRI engagement. They are called nonreactives.

Since Andosan^TM^ has an anti-inflammatory effect in blood donors and inflammatory bowel disease (IBD) patients [23,24,25,26] and aeroallergy and asthma can be defined as local inflammatory conditions in the airways, it was pertinent to investigate this mushroom extract in an allergy and asthma intervention study. We examined whether the AbM-based mushroom extract Andosan^TM^, taken orally before the pollen season, could reduce allergy and asthma symptoms, medication, and specific IgE basophil sensitization to t3 (birch pollen extract) and Bet v 1 (major component from birch) and could influence the Th1/Th2 cytokine balance in otherwise healthy humans as found in mouse models for allergy and asthma [9,10].

## 2. Materials and Methods

### 2.1. Trial Design

This is a single-center, randomized, two-armed, double-blinded study designed to determine whether daily oral intake of Andosan^TM^ mushroom extract before birch pollen season improved clinical symptoms and reduced medication and specific IgE and basophil sensitization in otherwise healthy blood donors during and after the study period. The donors were evaluated before, during, and after Andosan^TM^ or placebo consumption (60 mL once daily). This dose, but not 30 mL/day (unpublished), reduced levels of pro-inflammatory cytokines in healthy volunteers [23] and patients with IBD [24] or multiple myeloma undergoing high-dose chemotherapy with stem cell transplantation [27]. Treatment for 7 weeks in the latter gave no serious side effects.

### 2.2. Study Subjects

These were voluntary blood donors at Oslo Blood Bank (OBB), Oslo University Hospital (OUH), with self-reported birch pollen-induced allergy and/or asthma. Among 68 blood donors at OBB assessed for eligibility with self-reported birch pollen-induced allergy or asthma, 60 donors with positive specific IgE were recruited to the clinical trial through a letter. Criteria of exclusion were undergoing or planning to undergo allergen immune therapy (AIT), anti-IgE therapy, or cortisone injection. Twenty-nine donors were randomized stochastically to Andosan^TM^ intervention and 31 were randomized to placebo intervention (Figure 1). They were not travelling long distances during birch pollen season, giving all participants similar birch pollen exposure. Study groups were comparable, with insignificant differences with respect to details of demographic data and study subjects characteristics regarding allergy symptoms, comorbidity from other allergies and asthma, and medication for such conditions (Table 1).

Study subjects were recruited at OBB by the principal investigator (author F.M.) and three part-time nurses who handled information, study medicine, questionnaire, and blood sampling. Blood samples for IgE and cytokine measurements were taken from recruited blood donors in conjunction with blood donation (a) before the birch pollen season in Oslo 2016 (from end of April until mid-May), (b) during the whole pollen season (throughout August) (only IgE), and (c) after the season (September–October).

### 2.3. Sample Size

It was determined from a similar randomized clinical trial with Andosan^TM^ in 50 patients with ulcerative colitis or Crohn’s disease that 50 patients were needed for significant differences at the *p* = 0.05 level in symptoms and plasma cytokine levels [25,26].

### 2.4. Randomization: Sequence Generation and Type, Allocation Concealment Mechanism, and Implementation

Putative participants were numbered continuously and randomized stochastically (Microsoft Excel randomization table) by simple randomization into two treatment groups, Andosan^TM^ yes or no, thus coupling a number for each participant to a treatment before he/she signed up for the study. Time points for available interviews were listed in advance numerically, thus allocating study number. Study medicine (Andosan^TM^ or placebo) of similar color and odor was contained in similar plastic bottles marked with study numbers and given out to participants by study nurses. Only the principal investigator knew the study key. An independent colleague generated the allocation sequence, and the principal investigator enrolled participants and assigned participants to interventions.

### 2.5. Interventions—Study Medicine: Andosan^TM^ and Placebo Blinding

The mushroom extract Andosan^TM^ was provided by Immunopharma company AS (organization no. 994924273), Oslo, Norway, after import as mushroom juice from ACE company, Gifuken, Japan. The production (commercial information) process comprises fermentation and heat sterilization, conforming with Japanese health food regulations. Lipopolysaccharide content was <0.5 pg/mL (Limulus test, Chromogenix, Falmouth, MA, USA). Andosan^TM^ was imported to Norway as mushroom juice in accordance with Norwegian Food Safety Authorities and stored sterile at room temperature in dark, glass bottles until transfer/refill in larger 1.5-L plastic containers and storage at 4 °C until consumed (60 mL/day) by study participants—60 mL/75 kg is similar to the effective dose in allergic mice (200 μL/25 g) [10]. The placebo solution, kept in plastic containers and used similarly, was composed of a color-like drink with ionized water, salt, and caramel color (E150c; 0.5 mL/L). Study subjects, study nurses, data collectors, and outcome adjudicators were blinded.

### 2.6. Outcomes

Primary outcomes were analyzed by the following: (i) An in-house questionnaire for the blood bank setting modified from recommendations for clinical outcomes in AIT trials for allergic rhino-conjunctivitis [16] was given out at the beginning and collected at the end of study. It contained questions regarding specific symptoms from conjunctiva: itchy, red, and watery eyes; itchy, runny, and blocked nose; itchiness/irritation and blocking of the throat; blocking of the chest and dyspnea; and eczema and skin irritation. Each symptom was scored 1, with the total possible symptom score being 12. Additional scoring was based on the presence of general allergy symptoms (yes = 1, no = 0) and change from last season in general allergy symptoms (more = +1, unchanged = 0, less = −1). Medication was not scored but listed separately as frequency (per week) and types of medication (antihistamines, nasal corticoids, degranulation inhibitors, and asthma drugs) during the intervention period relative to before the intervention period.

(ii) Levels of total IgE (kU/L), IgE anti-Bet v 1 (recombinant birch allergen), and IgE anti-t3 (common silver birch extract) (kUA/L) were analyzed by ImmunoCAP^®^ technology in an ImmunoCAP^®^ 250 instrument (ImmunoDiagnostics, Thermo Fisher Scientific Inc., Uppsala, Sweden) in serum from blood sampled into glass tubes with gel or 15 IU heparin/mL for IgE and cytokine analyses, respectively. The levels of IgE anti-t3 (kU/L) analysed by Euroline atopy screen (IgE) test (Euroimmun AG, Lübeck, Germany) that gives similar results as ImmunoCAP^®^, was also examined in 50 male and 50 female unselected consecutive blood donors in mid-April of the previous year (2015).

(iii) BAT [28] for basophil sensitivity against t3 (common birch pollen extract) and rBet v 1 (gift from Euroimmun AG) allergens was tested by a FlowCAST^®^ kit (Bühlmann Laboratories AG, Schönenbuch, Switzerland) in whole EDTA blood according to manufacturer’s instructions. Basophils were identified by chemokine receptor type 3 (CCR3) [29] and their activation by CD63 surface expression by using a Gallios flow cytometer (Beckman Coulter, Brea, CA, USA) at 488 nm. Positive controls, anti-Fc EpsilonRI and *N*-formyl-methionyl-leucyl-phenylalanine, were used to detect nonreactives. When the positive control is negative, the test cannot be evaluated and such individuals are then called nonreactors. A patient individual base value of 2.0%–2.5% of the activated basophils was regarded as negative and used as the basis for determining allergen-specific cutoffs. An increase compared with negative control of <10% in CD63-positive basophils was considered negative, and an increase of >10% was considered a positive result. In vitro experiments were performed with 0–10% of Andosan^TM^ incubated with whole blood from untreated donors at 37 °C for 30 min and were subjected to BAT analysis.

The gating strategy for basophilic cells was performed according to the FlowCAST^®^ kit (Bühlmann Laboratories AG, Schönenbuch, Switzerland). Briefly, leukocyte population, which is separated into three discrete populations, was gated on a side scatter versus forward scatter (SSC-FSC) plot. The basophils are identified as CCR3^high^ SSC^low^ from the gated lymphocytes (SSC-CCR3-PE plot), and eosinophils located on the high right side have been excluded due to their SSC high position. Basophils are analyzed for activation (CD63^high^) in the CD63-FITC-CCR3-PE plot. The upregulation of the activation marker CD63 was calculated by the percentage of the CD63-positive cells compared to the total amount of basophilic cells (CCR3^high^ SSC^low^). In each assay, at least 200 basophils were assessed. We used stimulation buffer as the negative control and the two antibodies provided by the kit as positive controls.

Secondary outcomes were shown by cytokine levels in plasma before vs. after intervention. Multiplex bead-based sandwich immunoassay using Bio-Plex xMAP technology (Bio-Rad, Austin, TX, USA) was employed with Luminex IS 100 instrument (Bio-Rad Laboratories, Hercules, CA, USA), powered using Bio-Plex Manager (version 6.0.1) software (Bio-Rad Laboratories)) for analysis of 9 different cytokines (TNFα, IFNγ, IL-2, IL-4, IL-5, IL-10, IL-12p70, IL-13, and GM-SCF) following manufacturers’ instructions [30].

### 2.7. Study Subjects Flow, Losses, and Exclusions

Sixty specific IgE positive blood donors, 29 in the Andosan^TM^ and 31 in the placebo groups, were randomized for inclusion in this study (Figure 1). Six were excluded according to study protocol criteria, 1 and 2 in the respective groups took no study medicine, 1 in the Andosan^TM^ group missed attendance, 1 in the placebo group was hospitalized for severe allergy, and another with the above reference value (100 kUA/L) for specific IgE before pollen season was excluded from IgE evaluation, thus resulting in 27 participants in each group (Figure 1).

Intervention was from January 2016 until the beginning of birch pollen season (varying from mid-February until primo-April), and follow-up was during pollen season throughout August and, again after season, in September–October.

### 2.8. Age and Gender and Numbers Analyzed

Median age for the 54 included blood donors with birch pollen allergy was 39 years (range 20–64). There was similar number of men and women, ages, and additional allergies (approximately 75%) in the Andosan^TM^ and placebo groups (Table 1).

Subjects analyzed negative or below cutoff or lacking values before and during/after season were excluded from evaluation of outcomes of the particular analysis; for IgE analyses, 0 to 7 participants with negative or too low values and 5 to 8 with lacking values were excluded, and for cytokine analyses, a large proportion was excluded. For evaluation of basophil response, 1 non-reactor in each group was excluded and 2 and 7 non-reactives were respectively excluded in the Andosan^TM^ and placebo groups (Figure 1). Participants with negative IgE anti-Bet v 1 were also excluded from further evaluation of symptoms and medication score in the questionnaire. The number of participants then eligible for evaluation of analysis of total IgE, IgE anti-Bet v 1, and anti-t3 during the pollen season were 20 and 19, 18 and 14, and 19 and 15 for the Andosan^TM^ and placebo groups, respectively. After season, the respective numbers for total IgE were 16 and 18 and, for specific IgE, were 12 and 12. Participants eligible for pre- and post-seasonal comparison of BAT were 24 and 19 in the Andosan^TM^ and placebo groups, respectively.

### 2.9. Statistical Analysis

Data are presented as mean and standard error or as median and range values, depending on normal distribution. Parametric (*t*-test) and nonparametric (Mann–Whitney (M–W) and Wilcoxon rank sum and test) tests were used between study groups for normally or abnormally distributed data, respectively. Paired sample t-test and Wilcoxon signed rank test were used for within-group analysis and Pearson’s r or Spearman rank order correlation were used for correlations. All tests were two sided, and results were deemed to be statistically significant at *p* < 0.05. *p* values at 0.10 were considered to reflect a trend.

### 2.10. Registration and Protocol

The study was registered with unique protocol ID: ClinicalTrials.gov Identifier: NCT03198455. The authors confirm that all ongoing and related trials for this drug (food)/intervention are registered. See CONSORT Clinical Trials record website and CONSORT 2010 Checklist (Appendix A).

### 2.11. Statement of Ethics

The study was approved on May 28, 2015 by the regional ethics committee (REC—South East Norway, ref. 2015/716) and followed the guidelines of the Helsinki declaration. All study subjects were informed and signed a written consent for participation, including the option of study withdrawal. Blood donors were recruited and followed up at the OBB, OUH, Oslo, Norway, from November 2015 through October 2016.

## 3. Results

### 3.1. Symptom and Medication Score Reduction

Scores for 12 specific allergy-related symptoms from the questionnaire tended to be higher (*p* = 0.10) at inclusion in the Andosan^TM^ than placebo group (Table 1) but were equalized in the mid-intervention season (*p* = 0.97) (Table 2). Then, general allergy symptoms also tended to be relatively less (*p* = 0.10) in the Andosan^TM^ group. Moreover, when compared with the season prior (2015) to the intervention (2016), general allergy-related symptoms were significantly reduced (score: −1) in the Andosan^TM^ compared to the placebo group (*p* = 0.02) (Table 2) with no change (score: 0). However, there was no statistical difference between the groups in the specific symptom scores (2 and 1 less, respectively, *p* = 0.51). The included individuals with pollen allergies were allowed to use their regular medication, which was similar before intervention (Table 1). None-the-less, types of medication used was reduced in the Andosan^TM^ relative to placebo group (*p* = 0.03) and the frequency of medication showed a similar tendency (*p* = 0.11) (Table 2). During pollen season, there was no difference in the number of blood donations between the groups (*p* = 0.87).

At inclusion, 26% and 18.5% of participants suffered from pollen-induced asthma in the Andosan^TM^ and placebo groups, respectively (Table 1). Compared with the placebo group, Andosan^TM^-treated asthmatics experienced less asthma symptoms (median score: −3 (range: −5–0) for Andosan^TM^ vs. −1 (−3.5–0) for placebo; *p* = 0.01) and used fewer types of medication (−2 (−2–−1) vs. 0 (−0.5–0); *p* = 0.02) at lesser frequency (−7 (−14–−4) vs. 0 (0–0); *p* = 0.009) for their asthma and allergies during the intervention pollen season compared to the previous one.

### 3.2. Specific IgE Level Reduction

Before intervention, there were positive but low correlations between symptom score and total IgE (Spearman rank order correlation ρ = 0.31, *p* = 0.02), IgE anti-t3 (ρ = 0.30, *p* = 0.02), and IgE anti-Bet v 1 (ρ = 0.25, *p* = 0.06). Changes in specific IgE anti-Bet v 1 levels during the pollen season relative to before were significantly reduced in the Andosan^TM^ compared to the placebo group (Table 3; *p* = 0.007). There was a similar finding for IgE anti-t3 levels (*p* = 0.01) but not for total IgE (*p* = 0.14; Table 3). The reduction in relative values (indices) during season in the Andosan^TM^ vs. placebo group was 36%–37% for specific IgE anti-Bet v 1 and anti-t3, 16% for total IgE, and still 22% (indices 0.97 vs. 1.25) after season for IgE anti-Bet v 1 (Table 3). The current selected blood donors with self-reported pollen allergy and specific IgE-verified sensitization (placebo group examined) had significantly higher IgE anti-t3 levels than 100 unselected donors (median 10.2 vs. 0.0 kU/L, respectively; *p* < 0.001) that were examined the previous pollen season (mid-April 2015).

### 3.3. Basophil Sensitization Reduction

Blood samples for basophil activation were taken before and after pollen season, together with samples for IgE and cytokine measurements, and determined by CD63 expression by flow cytometry. Those allocated to placebo treatment tended to have basophils that were less sensitive to pollen activation (*p* = 0.062) (Table 4). In the placebo group, the maximal (peak) allergen concentrations needed for eliciting basophil activation before the season mostly changed to lower peak concentrations after the season (*p* = 0.004; Table 4; Figure 2a). In contrast, for patients with allergies allocated to Andosan^TM^ treatment, most BAT peak profiles did not change from before to after the pollen season (*p* = 0.312; Table 4; Figure 2b). When comparing the resultant changes observed during the season within each group, there was a relative and significant (*p* = 0.028) shift in the Andosan^TM^ group to less changes in allergen sensitivity (Table 5; Figure 2c). In fact, whereas the mean peak Bet v 1 concentration for basophil activation was reduced 10-fold in the placebo group, it was doubled in the Andosan^TM^ group (Table 4). Hence, Andosan^TM^ treatment before the pollen season rendered basophils less sensitive to allergen activation during the season as compared with placebo.

When participants whose blood samples were analyzed after the tree pollen season (August) were separated from those analyzed later (September–October), there were similar findings as above compared with pre-seasonal BAT measurements for both cohorts, a significantly lower activation in the Andosan^TM^ (change in peak Bet v 1 conc. needed: 0 (−0.25–0)) than the placebo group (−1 (−2–0)) (*p* = 0.046). Also, for participants examined for basophil activation later in September–October, there tended to be similar findings in favor of Andosan^TM^ treatment (*p* = 0.06).

### 3.4. Basophil Activation: IgE and Symptoms (Allover Allergy Ailments) Negative Correlations

There were negative and significant correlations between basophil activation (Cmax) values and IgE anti-Bet v1 (Pearson’s r −0.43, *p* = 0.006), IgE anti-t3 (correlation coefficient (corr. Coeff.): −0.44, *p* = 0.004), and total IgE (corr coeff −0.34, *p* = 0.032). This agrees with basophils being activated by less allergens (becoming more sensitized) in people with allergies with high specific IgE values and vice versa. Also, for changes in basophil activation values and symptoms, there was a significant negative correlation (*p* = 0.015) (corr coeff −0.38).

### 3.5. Basophils Are Not Activated In Vitro by Andosan^TM^ Stimulation

When basophil activation was analyzed before participants entered the study, there was no difference in BAT between basophils pre-incubated in vitro with Andosan^TM^ (1%–10%) or saline (*p* > 0.05), indicating no direct effect of Andosan^TM^ on basophils.

### 3.6. Plasma Cytokines Changes Not Measurable

When cytokines TNFα, IFNγ, IL-2, IL-4, IL-5, IL-10, IL-12p70, IL-13, and GM-SCF were measured in plasma and examined before and after intervention in the two groups, there were either no differences or the levels were too low (0 or below lowest standard value) to be analyzed.

### 3.7. No Harm

One participant in the Andosan^TM^ and two participants in the placebo groups did not take the study medicine because of taste. Otherwise, no harm was recorded.

## 4. Discussion

This randomized, double-blinded, and placebo-controlled study in birch-allergic blood donors suggests that add-on Andosan^TM^ pretreatment before birch pollen season generally improves allergy ailments—although not shown by specific symptom scores—and reduces need for medication. One major challenge in this study was to see the effects of Andosan^TM^ supplementation on top of regular drugs taken for allergy and asthma. The different results regarding changes from previous season between twelve specific allergy symptoms and general allergy symptoms may reflect that memory of specific symptoms the year before is less precise than global memory of change in allergy-related quality of life.

The clinical outcome agrees with preclinical findings showing that *Agaricus blazei* extracts protected against allergen sensitization and asthma in mice [9,10]. Based on the positive albeit low correlation between specific IgE and allergy symptoms before intervention, the close to 40% reduction in specific IgE levels in the Andosan^TM^ group during pollen season is assumingly associated with the clinical findings. This is supported by the negative correlation between change in BAT resultsand change in allergic ailments, indicating that when basophils became less sensitized to allergen by Andosan^TM^ supplementation, the symptoms decreased. However, it could not be explained by alterations in plasma cytokines that were difficult to measure. 

Plasma cytokine levels were either too low to measure or there were no changes before compared to after the season, when the cytokine concentrations most probably had reverted back to pre-seasonal levels. Preferably, cytokines should have been measured again during the season to see putative changes induced by Andosan^TM^ intervention as seen for change in specific IgE (Table 3). Although Andosan^TM^ given orally has an effect on cytokines in plasma [11,23,24,26,27], plasma samples are not ideal for analyses of putative topical changes in cytokines caused by local inflammations, e.g., in asthma and aeroallergy, but should rather be measured locally in bronchial aspirates and nasal secrete/tears, respectively. In the mouse allergy model, amelioration of the Th1/Th2 imbalance was found in spleen cells harvested from animals treated with an AbM extract or Andosan^TM^ [9,10]. Andosan^TM^ also had anti-inflammatory effects both in blood donors and IBD patients [23,24].

After the pollen season, specific IgE levels in the Andosan^TM^ group rose again towards pre-seasonal levels (Table 3), suggesting that the extract did not affect IgE beyond the season but stabilized IgE levels that normally peak mid-season. However, the finding that basophil activation, in contrast to specific IgE levels, was reduced significantly after the pollen season suggests that the Andosan^TM^ effect on basophil sensitization lasted longer. The density of expressed basophil FcεRI correlates with levels of serum IgE, which stabilizes the receptor at the cell surface [17]. Hence, the lower circulating specific IgE during pollen season in Andosan^TM^-treated individuals likely rendered their basophils less sensitive to allergen cross-binding and activation by reducing specific IgE binding to FcεRI on the cells. The negative correlations between changes in BAT results and change in specific and total IgE levels agrees with the notion that decreas in IgE levels during pollen season stabilizes less FcεRI on basophils and thus decreases the cells’ sensitivity to pollen allergen.

Presumably, in vivo Andosan^TM^ supplementation for people with allergies had a similar indirect effect on mast cells as on basophils because an *Agaricus blazei* extract has been shown to have both anti-inflammatory and anti-allergic effects in bone marrow-derived mast cells in vitro [31]. These effects were suggested to occur through blocking of COX-2 expression and protein kinase B activation, which are associated with release of inflammatory and allergic mediators in mast cells such as prostaglandin D_2_ and leukotriene C_4_ [32].

The preferred method for evaluating basophil sensitivity to allergen is calculation of 50% of the effective concentration (EC50) [22]. However, since the needed wide variation of eight declining 10-fold concentrations to calculate EC50 was missing in most patients, peak shift within three overlapping concentrations between analyses before and after intervention in each group (A or P) was used instead. This is generally accepted in BAT examinations [22]. When comparing BAT peaks before intervention in the two groups, there was no statistical difference but there was a tendency (*p* = 0.062) to be less sensitized for participants in the placebo than Andosan^TM^ group. Despite the randomization, this suggests that those allocated to later Andosan^TM^ supplementation tended to be more allergy prone during the pollen season.

Based on improved Th1/Th2 balance, most studies seem to agree that β-glucan has a potential as adjunct treatment of patients with allergies [33]. However, since the β-glucan content of Andosan^TM^ is low [34], the biological properties of Andosan^TM^ must be attributed also to other glycans detected, which inhibited activity of the proinflammatory and tumor-associated endopepetidase legumain in macrophages [34]. Legumain may play a role in atopic dermatitis [35], and there was reduced expression of this enzyme in intestines of mice given Andosan^TM^ orally [11]. In addition, the mushroom extract contains low protein but higher lipid concentrations (not shown) that may explain some of its effects.

How Andosan^TM^ interacts with antihistamines, corticosteroids, and other drugs for allergy and asthma is unknown in contrast to its interference with cytochrome P-450 metabolism and transmembrane efflux pump P-glycoprotein (P-gp), which governs pharmacokinetics and drug concentrations [36]. The AbM extract Agaricus Gold Label from Japan, later named Andosan^TM^, inhibited P-gp transport and cytochrome P450 (CYP3A4) metabolism in similar or lower concentrations than did green tea [37,38], indicating that clinical or intestinal interactions of *Agaricus* with CYP3A4 were unlikely [38]. This agrees with the lack of side effects observed in clinical trials with IBD [25] and multiple myeloma patients [27] after Andosan^TM^ supplementation to regular treatments for 3–7 weeks. 

Andosan^TM^ probably has effects against various allergens because of its nonspecific anti-allergic effect as shown against both food- (ovallergen) [10] and birch pollen-induced sensitization. Since inhibitory effects of AbM has been shown on mast cell-mediated anaphylaxis reactions in mice [39], Andosan^TM^ may also exhibit similar effects. Another interesting aspect is whether Andosan^TM^ could be used as an adjuvant for AIT similar to experimental use of AbM and β-glucan in vaccines for viral infections in animal models [40,41]. Since the extract also had antibacterial effects in murine sepsis models [42,43], it would have been interesting to examine the anti-allergic effects of Andosan^TM^ in patients with combined allergic and infectious conditions as often seen in asthma/bronchiolitis and atopic dermatitis in infants and children.

The protection observed against basophil activation in pollen-allergic patients after Andosan^TM^ supplementation before pollen season seems to taper off after August. At this time, the season is long gone for birch pollen and mostly over for grass pollen in southern Norway. This suggests that a prophylactic regimen similar to the current one with this mushroom extract could reduce pollen-induced allergy problems. The supplementation most probably has to be repeated to have positive effects in the subsequent pollen season, but this was not examined. Also, we do not know whether the anti-allergy effect of Andosan^TM^ will be diminished over time if people with allergies develop tolerance to it. Further studies are warranted for reaffirmation of the present results.

## 5. Conclusions

This randomized placebo-controlled clinical study suggests that pre-seasonal oral supplementation with Andosan^TM^ mushroom extract can have anti-allergic effects in pollen-induced allergy by protecting against basophil sensitization during pollen season. Most probably, prevented mastocyte sensitization, albeit not examined, is the main mechanism behind the reduction in general allergy symptoms and medication in the participants of this study.

## Figures and Tables

**Figure 1 nutrients-11-02339-f001:**
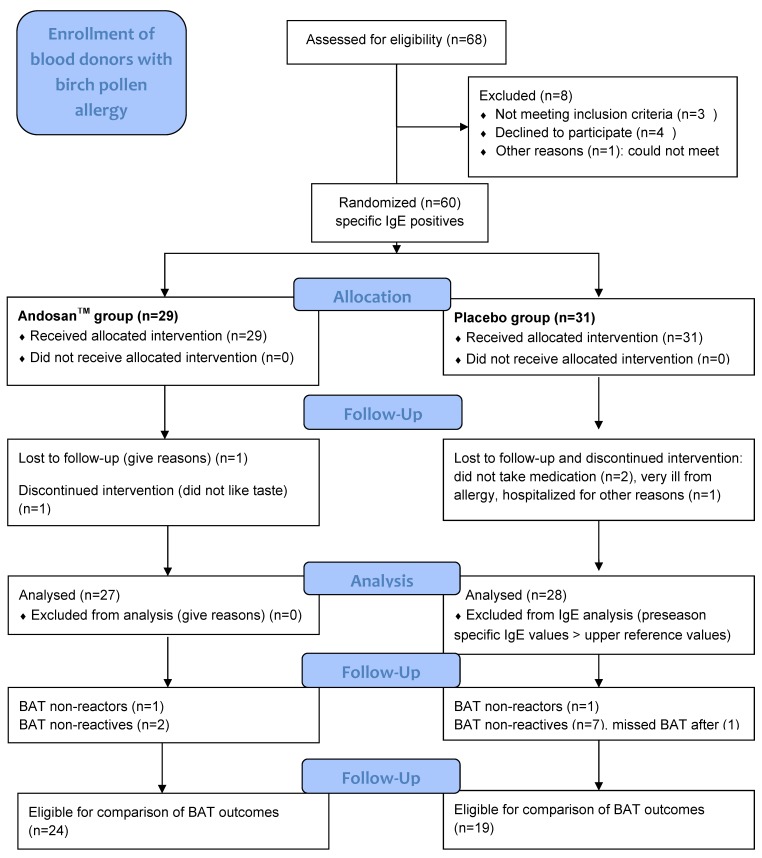
CONSORT (Consolidated Standards of Reporting Trials) 2010 flow diagram. Definitions: specific IgE (IgE anti-Bet v 1), BAT (Basophil Activation Test).

**Figure 2 nutrients-11-02339-f002:**
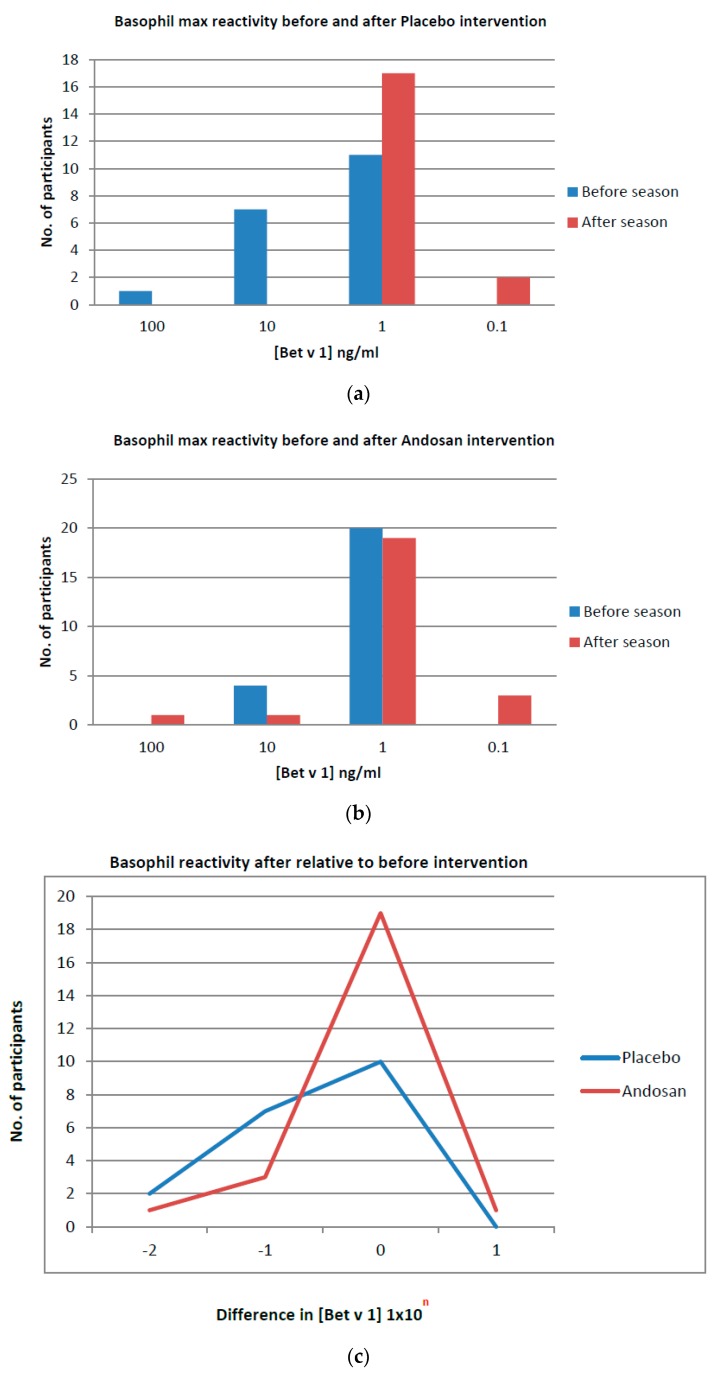
Basophil reactivity in the intervention groups. Basophil reactivity is shown before and after Placebo (**a**) or Andosan^TM^ (**b**) intervention, and after relative to before the respective interventions (**c**). Also see Table 4 for depicted data in (**a**,**b**), and Table 5 for (**c**).

**Table 1 nutrients-11-02339-t001:** Demographic and blood donor data.

	Andosan^™^	Placebo	*p*-Value
Number	27	27	-
Age (years)	40 range 22–60	37 range 20–64	0.62
Gender (male, female)	15, 12	14, 13	-
Specific symptoms, pollen allergy	6 range 4–7	5 range 3–6	0.10
Comorbidity: allergy other than for birch	20	21	-
Comorbidity: asthma	7	5	-
Medication: types of allergy medication	1 range 1–3	3 range 1–3	0.14
Medication: frequency of medication during season	7 range 7–14	7 range 7–14	0.42

Values for age and medication are given as number or median (range 25%–75%). (Statistical analysis: Mann–Whitney (M–W) rank sum test). The table describes participants included for analysis.

**Table 2 nutrients-11-02339-t002:** Intervention data: allergy.

	Andosan™	Placebo	*p*-Value
Allergy symptoms			
General allergy symptoms mid-season	0 (0–1)	1 (0–1)	0.10
Specific allergy symptoms mid-season	5.5 (4–7.7)	5.0 (3–7)	0.97
Change in general allergy symptoms (2016 vs. 2015)	−1 (−1–0)	0 (−1–0)	0.02
Change in specific allergy symptoms (2016 vs. 2015)	−2 (−4–0)	−1 (−3–0)	0.51
Allergy medication			
Types of drugs (antihistamines, nasal corticoids, and degranulation inhibitors)	1 (1–1)	1 (1–3)	0.03
Frequency of medication	7 (3–7)	7 (1–14)	0.11
Number of blood donations April–August	1 (0.5–1)	1 (0–1)	0.87

Values are given as median (range 25%–75%). (M–W rank sum test). Scores: General allergy symptoms (yes = 1, no = 0), change in general allergy (more = +1, no = 0, less = −1); specific symptoms total score (*n* = 12, 1 or 0 for each), and change = difference in the total score. The table describes results from participants included for analysis.

**Table 3 nutrients-11-02339-t003:** Intervention data: IgE measurements.

IgE Ab	Indices	*n*	Andosan^TM^	*n*	Placebo	*p*-Value
Total IgE	During/Before	20	0.93 (0.81–1.06)	19	1.11 (0.82–1.55)	0.14
	After/Before	16	1.02 (1.86–1.21)	18	1.31 (0.98–1.62)	0.14
Bet v 1	During/Before	18	0.82 (0.72–1.15)	14	1.30 (0.94–2.14)	0.007
	After/Before	12	0.97 (0.81–1.31)	12	1.25 (0.92–1.45)	0.21
t3	During/Before	19	0.86 (0.76–1.07)	15	1.35 (0.95–2.33)	0.01
	After/Before	12	1.02 (0.82–1.24)	12	1.14 (0.80–1.50)	0.46

Wilcoxon signed rank test was used. Levels before pollen season were for total immunoglobulin E (IgE): mean 107.7 ± 25.6 standard error of the mean (SEM)/ median 61 (range 2–892); for Bet v 1: 10.3 ± 1.6/6.4 (0–37); and for t3: 10.3 ± 1.6/6.6 (0–36) kUA/L. Participants with negative (*n* = 0–7) or missing (*n* = 5–8) values were excluded. The indices represent IgE levels during or after the pollen season divided by the respective IgE levels before the season. Test for difference in Andosan^TM^ before (median 9.2) vs. placebo before (median 8.9): *p* = 0.85 (M–W rank sum test).

**Table 4 nutrients-11-02339-t004:** Comparisons before and after intervention of maximal (peak) Bet v 1 concentration needed for basophil activation.

Bet v 1 conc.	Placebo		Andosan	
	Peak before	Peak after	Peak before	Peak after
	*n* (%)			
100	1 (5.2)			1 (4.1)
10	7 (36.8)		4 (16.6)	1 (4.1)
1	11 (57.8)	17 (89.4)	20 (83.3)	19 (79.1)
0.1		2 (10.5)		3 (12.5)
Mean ng/mL	9.52 ± 5.12	0.90 ± 0.06	2.50 ± 0.69	5.38 ± 4.12
Total	19 (100)	19 (100)	24 (100)	24 (100)
Test for difference	*p* = 0.004		*p* = 0.312	

Test for difference: *p* = 0.004, *p* = 0.312 (Wilcoxon signed rank test). Test for difference in placebo before vs. Andosan^TM^ before intervention: *p* = 0.062. (M–W rank sum test). Values represent number of individuals and percentages of total. Concentration (conc.).

**Table 5 nutrients-11-02339-t005:** Shift in maximal Bet v 1 concentration needed for basophil activation from before until after intervention.

Diff in (Bet v 1) 1 × 10^n^	Placebo	Andosan
**logarithmic Power**	***n***	
−2	2	1
−1	7	3
0	10	19
1	0	1
Total	19	24
Test for difference	*p* = 0.028	

Test for difference: *p* = 0.028 (M–W rank sum test). Values represent number of individuals.

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
