# Peer review of "Agaricus blazei-Based Mushroom Extract Supplementation to Birch Allergic Blood Donors: A Randomized Clinical Trial"

_nutrients, 2019, doi:10.3390/nu11102339_

Round 1

Reviewer 1 Report

--GENERAL OVERVIEW--

The submitted manuscript ‘Nutrients-573459’ presents the results of a controlled study on prophylactic effects of oral Agaricus blazei Murill (AbM) extract in subjects with birch allergy.  Results described that the treatment group had a reduction in reported general allergy symptoms, reduced specific IgE antibodies against birch allergens, and higher concentration of birch allergen required to activate basophils.  The only major comment I outline below, under line 31, is to avoid over-concluding about asthma.  The introduction of the paper does a very nice job explaining the mechanism of type I allergy, and the data are mostly relevant to that condition.

--SPECIFIC COMMENTS--

ABSTRACT:

Line 24 – Since the “general allergy and asthma symptoms” were reported and not measured, I think it’s important to add the word ‘reported’ before those words.

Line 25 – If there’s room, you could specify that "t3" is (birch pollen extract)

Line 31 – The self-reported asthma data is acceptable to leave in the Results section.  However, in my opinion, the word “asthma” should be removed from the conclusions in lines 31, 329 and 432 because clinical asthma outcomes were not additionally measured.  Asthma is a more serious condition than birch pollen allergy, and therefore a higher burden of proof would be required to conclude on effectiveness for that condition.  The birch allergy conclusion is acceptable because clinical markers of allergy were measured in addition to the patient recall.  Also, please consider changing the wording from “must be due to”, to, ‘was associated with’.

RESULTS

Line 258 – The meaning of the word “under” is not clear to me.  Do you mean ‘during’ the season.  Please adjust it on Table 3 as well.

Line 312 – The line says, “Basophil activation” was negatively correlated with symptoms.  Please add the word ‘values’, or something like it, because in its absence the opposite meaning is conveyed.  This is also the case in lines 340 and 374, where the words “outcome” and “values” could be replaced with something like, ‘elicitation concentration’.

Line 316 – Where the in vitro basophil culture is described, please specify the concentration of AbM that was used.

DISCUSSION

Line 372 – This sentence ends with, “reducing specific IgE binding to FcεRI on the cells”.  Please add the word ‘likely’…“rendered their basophils…”.  FcεRI data isn’t shown and therefore it can’t be absolutely concluded that this was an outcome.  Similarly in lines 375-376, you could remove the words “stabilizes less FcεRI on basophils and thus...”.FIGURES AND TABLES

Figure 1 – There appears to be missing text in the box: “Analysed (n=28); Excluded from IgE analysis (preseason”

Tables 1 and 2 – Please add text to indicate to the reader at what point in the study each table represents; either to the table legend or to the relevant rows.

Table 4 vs Figures 2a-b.  These appear to present the same data in two different formats.  It depends if the journal is accepting of that, but you might consider choosing one of these formats and removing the other.

Table 5 vs. Figure 2c.  Again, these appear to present the same data.  Therefore you may consider using only one of these formats.  Also, Figure 2c is lacking an x-axis label: Difference in [Bet v1] 1x10n (logarithmic power).

Thank you for the opportunity to review the manuscript.

Author Response

GENERAL OVERVIEW - major comment: Over-conclusions about asthma are now removed.

SPECIFIC COMMENTS: Changes has been done in the TEXT according to reviewer's request, please se in revised manuscript the new Lines 24 ("reported"),  26 ("(birch pollen extract)"), 31, 341, 445 and 447  ("asthma" omitted), 32 ("was associated with"),  265, 275 and Table 3 ("during"), 324 ("values"), 352 and 386 ("elicitation"), 328 ("AbM concentration" : 1-10%), 384 ("likely"), 388 (" stabilizes less ..." removed). 

FIGURES AND TABLES

Figure 1 - missing text in boxes is now included

Tables 1 and 2: Point in the study that each table represents, is included in new lines under each table ("The table describes  ...").

Table 4 vs Figures 2a-b: I think both are nedded because the table also gives mean values and shows the statistical difference between the interventions.

Table 5 vs. Figure 2c: Table 5 shows the data basis for Figure 2c, in addition to the test for difference. on Figure 2c x-axis "logarithmic power" is included.

Reviewer 2 Report

GENERAL COMMENTS

This paper completes the studies that have been carried out so far on the properties of the AndosanTM extract, now on the perspective of birch pollen allergies and asthma. The study is interesting although the results would need to be completed with subsequent studies that reaffirm them. The authors should follow the suggestions given below:

INTRODUCCTION

Lines 49- 51. The authors should enter references that support the statement made in these lines.

Line 59-60. The authors should enter references that support the statement made in the sentence between these two lines.

RESULTS

The authors should move paragraphs 3.1 and 3.2 from the Result section to Material and Method section.

Lines 258-259. The authors should explain in more detail the results shown in this sentence in relation to the values shown in Table 3. The authors do not show the total values of the variables studied during and after the treatment (only before) and therefore the percentages indicated in the text are not well explained.

In Table 3 the results with statistical significant differences refer to the relationship between AndosanTM versus Placebo treatments, before and during the pollen season. However, in the Result section the authors allude to a 22% reduction in AndosanTM treatments after the pollen season for IgE anti-Bet v 1. The authors have to show and explain the data that support this statement.

Line 288. In this line the authors indicate that they do not show the data (data not shown). These data should be shown because they are important to support the discussion of this work.

DISCUSSION

The authors state in the Discussion section that pre-treatment with AndosanTM before birch pollen season generally reduces asthma symptoms. However, the results do not show the data that support this statement (they only show the values of p) and these data should therefore be attached.

Lines 353 to 365 and 418 to 420. The authors explain in the text contained in these lines the properties of AndosanTM already published in other works without directly relating them to the results of this work and therefore these lines can be eliminated.

The authors should add that this work would need to be completed with subsequent studies that reaffirm the results obtained.

FIGURES

Figure 1. The authors should explain why in the placebo group 28 sera from patients were analyzed and were only eligible 19. The authors must include that the BAT non-reactive sera are n = 7

Figure 2a. The information contained in this figure is also contained in Table 4. Therefore this figure is redundant and can be deleted.

Figure 2. The information contained in this figure is also contained in Table 5. Therefore this figure is redundant and can be deleted.

TABLES

Table 1. The authors should specify the meaning of the letter “P” in the legend

Table 2. The authors should explain that the values that appear for “Change in general allergy” and “Change in specific allergy symptoms” refer to the data for 2016 in relation to 2015.

Table 4. The authors should add in the legend of the table that the values represent the number of individuals and the percentage with respect to the total.

Author Response

Author’s replay to Reviewer 2, first report:

INTRODUCTION

Lines 48-51: New refs # 12 and 13 are given.

Line 59-60: Reference is given to in-house questionnaire.

RESULTS

The paragraphs have been moved from Results to Materials and Methods.

Lines 258-259: More explanatory details are now given.

Now I explain data supporting the alluded 22% reduction shown in Table 3.

Line 288: Data are now shown in parenthesis in text.

DISCUSSION

Data for reduction in asthma symptoms and medication is now attached in the “Asthma” paragraph in Results.

Lines 353-365 and 418-420 have been eliminated together with references that only occur there.

Statement about further studies being warranted for reaffirmation of the results is attached at end of Discussion.

FIGURES

Figure 1. All text in the figure boxes is now shown, which explains the difference between 28 at start and only 19 being eligible. (The number 28 refers to the number after exclusion of 1 with pre-seasonal too high value).

Figure 2a, 2b and 2c are now removed.

TABLES

Table 1. “P” is replaced by P-value also in Tables 2 and 3.

Table 2. “Change in general allergy” and “Change in specific allergy symptoms” is followed by reference of 2016 vs 2015.

Table 4. A sentence regarding the values representing number of individuals and percentages of total, is added.

Reviewer 3 Report

Authors demonstrated that Agaricus blazei Murill (AbM) extract could reduce general allergy and asthma symptoms and consumption of medications in individuals with self-reported birch pollen allergy and/or asthma. AbM reduced also serum specific IgE (sIgE) level and maintained low level of basophils response to allergen in BAT.

Apart from clinical outcomes, the main part of analysis refers to sIgE and BAT results. However, author didn’t refer in the discussion to any results from studies in similar human model (I mean - effect of treatment with common drugs on sIgE or BAT results in patients with allergic rhinitis). If there are such studies – similar results can support findings in the paper. If there are no analogous studies – it increases originality of the paper.

Authors stated in conclusions: “AndosanTM mushroom extract can have anti-allergic effects in pollen-induced allergy and asthma by protecting against basophil sensitization during the pollen season. Most  probably, the prevented basophil sensitization is the main mechanism behind the reduction in general allergy and asthma symptoms and medication in the participants of this study.” Mastocytes in mucosa are main effector cells, basophiles often substitute them in experiments, but these are not the same cells; asthma is more complicated than histamine release from mastocytes in bronchi only (and we do not know how many asthmatics were evaluated); main mechanism is also more complicated, here rather effector mechanism can be considered. Taken together, conclusions are overestimated and speculative a little bit. Say what you find, no less, no more.

Minor comments:

- Category “Types of drugs” (Tab. 2) seems to me a bit artificial/ laconic, when one analyze the results of intervention. If it is impossible to show differences in particular drug categories (e.g. antihistamines, nasal corticosteroids, anti-asthmatic?), please enumerate groups of drugs which have been included into analysis, at least.

- Authors use terms “non-reactor”, “non-reactives”, and “non-responders” in the text and in Fig. 1. What is the difference? Please, clarify or unify terminology (“non-responders” is used in papers on BAT)

- Fig. 1. could be edited more carefully, and it seems to me something is missing in the right arm (placebo)

- Fig. 2a, Fig. 2b – What about a legend for a red columns?

Author Response

About reference to results from similar studies: It is referred to two studies (ref # 12,13) in Introduction, new Lines 54 and 55 in which β-glucans were used against allergic rhinitis.

Also in Materials and Methods Lines 105-106 in revised manuscript, reference is given to allergen immune therapy (AIT), anti-IgE therapy and cortisone injections that are common but cumbersome and expensive (AIT and anti-IgE) or temporary and with potential side-effects (cortisone) treatments for allergic rhinoconjunctivitis today.

Conclusions regading asthma has been changed according to reviewer's request; Lines 31, 341, 445, 447.

Minor comments: "Types of drugs" are now included as drug categories (antihistamines, nasal corticosteroids and degranulation inhibitors) in Line 145 and in Table 2.

"Non-reactives" and "non-reactors" are now defined in Lines 81, 160-161, where also "non-reactives" is used instead of "non-responders", which is a wrongful term.

Figure 1: all text in boxes is now included. 

Figure 2a and 2b now has legend also for red columns.

Round 2

Reviewer 2 Report

The corrections made by the authors are appropriate and the new version of the paper has improved the previous one.
